# Natural Products Induce Lysosomal Membrane Permeabilization as an Anticancer Strategy

**DOI:** 10.3390/medicines8110069

**Published:** 2021-11-10

**Authors:** Reginald Halaby

**Affiliations:** Department of Biology, Montclair State University, Montclair, NJ 07043, USA; halabyr@montclair.edu

**Keywords:** cancer, lysosomes, lysosomal membrane permeabilization, apoptosis

## Abstract

Cancer is a global health and economic issue. The majority of anticancer therapies become ineffective due to frequent genomic turnover and chemoresistance. Furthermore, chemotherapy and radiation are non-specific, killing all rapidly dividing cells including healthy cells. In this review, we examine the ability of some natural products to induce lysosomal-mediated cell death in neoplastic cells as a way to kill them more specifically than conventional therapies. This list is by no means exhaustive. We postulate mechanisms to explain lysosomal membrane permeabilization and its role in triggering cell death in cancer cells.

## 1. Introduction

Cancer is the second leading cause of death worldwide. Many tumors eventually become resistant to hormones, chemotherapy, and radiation by avoiding apoptosis [1]. Cancer cells become resistant to anticancer therapies by mutating pro-apoptotic genes and upregulating anti-apoptotic genes. This chemoresistance is one of the biggest reasons why chemotherapeutic therapies fail. Treatments such as chemotherapy and radiation are known to have untoward side effects. The use of natural products may have fewer side effects and less toxicity than conventional chemotherapeutic drugs [2]. The chemoprotective properties and low cytotoxicity of natural products make them attractive resources to use against malignancies. It is desirable to identify more natural compounds with anticancer activity. The antineoplastic actions of these natural compounds are mediated by their ability to induce apoptosis in cancer cells [3]. Apoptosis or programmed cell death is regulated by a balance of activation of proapoptotic genes, such as executioner caspases (including caspases −3, −6, and −7 [4,5]) and antiapoptotic genes, such as Bcl-2 and XIAP. This review will focus on the ability of certain natural products to induce apoptosis by triggering lysosomal membrane permeability (LMP). Lysosomal-mediated apoptosis is an attractive way to target neoplastic cells since cancer cells have larger and thus more fragile lysosomes compared to wild-type cells [6,7]. Moreover, cancer cells have a higher reliance on lysosomes for proliferation, metabolism, and adaptation to stressful environments relative to normal cells. Indeed, cancer cells can increase the biogenesis of lysosomes, thus affecting the number of lysosomes [8,9]. Unlike mutating genes, neoplastic cells cannot alter their lysosomes, rendering these organelles as putative sites for directing novel anticancer treatments. Thus, lysosomal cell death offers an alternative mechanism to kill tumor cells that become resistant to standard chemotherapy. Lysosomes have been reported to play a role in sequestering basic chemotherapeutic drugs in their acidic lumens, thus lowering the effective drug concentrations to target sites, such as the nucleus [10,11]. Clearly, further investigations are warranted to decipher the exact roles played by lysosomes in cancer therapy.

Lysosomes are acidic organelles that contain at least fifty hydrolytic enzymes including proteases, nucleases, glycosidases, and lipases [12]. Lysosomes digest unwanted materials and damaged organelles. These hydrolases can degrade the entire contents of a cell, which is why they must perform cellular digestion within the lysosomal membrane. Leakage of lysosomal enzymes into the cytosol can initiate apoptosis [13,14,15]. Furthermore, cleavage of Bid and degradation of Bcl-2 by lysosomal cathepsins can promote mitochondrial membrane permeabilization and caspase activation, which are hallmarks of apoptosis [16,17]. Lysosomal hydrolases can also initiate the intrinsic apoptotic pathway independent of Bid cleavage [15]. The most relevant lysosomal proteases are cathepsins B, D, and L; they are abundant in lysosomes and can remain active at neutral pH values [16,18]. The intrinsic (mitochondrial) apoptotic pathway involves the release of intermembrane space proteins such as cytochrome c and Smac/DIABLO and activation of executioner caspases [19,20,21]. This review will focus on the ability of certain natural products to induce apoptosis by triggering lysosomal membrane permeability (LMP).

## 2. Lysosomal Membrane Permeabilization

Mounting evidence suggests that lysosomes are good molecular targets for cancer therapy [22,23,24]. The cytosolic translocation of lysosomal enzymes can be triggered by reactive oxygen species, lysosomotropic agents, and weak bases, including chemotherapeutic agents [25,26]. A recent study identified autophazole, a novel autophagy initiator that gets incorporated into lysosomes of cancer cells [27]. Autophazole induces the release of cathepsins from lysosomes, leading to apoptosis. Some anticancer agents are known to induce lysosomal-dependent cell death by modifying the lysosomal membrane integrity, including vincristine and siramisene [28,29].

The precise mechanisms responsible for regulating lysosomal membrane permeabilization (LMP) have yet to be elucidated. It is not known whether pores or channels form in the lysosomes. It has been confirmed that the following agents and signaling pathways can disrupt lysosomal membranes, namely reactive oxygen species [30], sphingosine [31], downregulation of Hsp70 [32], photodynamic therapy [33], and translocation of Bax into the lysosomal lumen [34]. Upon release into the cytosol via LMP, cathepsins degrade Bcl-2 and cleave Bid, triggering the mitochondrial apoptotic pathway [35]. Regardless of the trigger of LMP, it has been shown by several reports that cytosolic leakage of cathepsins precedes changes in the mitochondrial membrane potential [18,36].

Possible explanations for the control of LMP are emerging. One report confirmed the occurrence of LMP via galectin 3 puncta assay as well as cytoplasmic leakage of lysosomal enzymes [37]. A recent study provides a putative explanation for the regulation of LMP. Toll-like receptor 3 (TLR3) acts as a death receptor in several cancer cell lines [38,39]. TLR3 can activate the extrinsic apoptotic pathway via initiator Caspase-8 [40,41]. Caspase-8 can then subsequently activate downstream effector caspases such as Caspase-3 and trigger the intrinsic apoptotic pathway. Loquet et al. (2021) showed that cell lines deficient in Caspase-8 undergo an unconventional type of cell death characterized by permeabilization of the lysosomal membrane as the initial event [42]. Interestingly, TLR3 is localized in lysosomes [42] and might provide a way to execute LMP in cancer cells that are defective in Caspase-8 or perhaps independent of Caspase-8.

## 3. Natural Products Induce Lysosomal Membrane Permeabilization in Cancer Cells

Several natural products have been identified that kill cancer cells by activating LMP, see Table 1.

Venkatesan et al. reported that Pinus radiata bark extract (PRE) induces apoptosis in MCF-7 breast cancer cells [43]. This group demonstrated that PRE-induced cell death was accompanied by lysosomal membrane permeability and concomitant cytosolic release of cathepsins. Furthermore, this cell death was observed to be caspase-independent. Although this cell death did not involve caspase activation, it possessed certain hallmarks of apoptosis. Namely, externalization of phosphatidylserine, cytoplasmic vacuolation, and chromatin condensation were observed.

Oleocanthal-rich compounds such as olive oil have been demonstrated to induce cell death in various cancer cells [44]. Moss et al. showed that low density lipoproteins reconstituted with the natural omega 3 fatty acid docosahexaenoic acid [45] (LDL-DHA) were selectively toxic to liver cancer cells and not normal hepatocytes [46]. This study demonstrated that basal levels of oxidative stress were higher in malignant TIB-75 cells compared to normal TIB-73 cells. The increase in reactive oxygen species (ROS) and iron-catalyzed reactions made cancerous liver cells susceptible to destabilization of their lysosomes. Another report demonstrated that DHA-treated cells induced lysosomal-mediated cell death in MDA-MB-231 breast cancer cells [47].

Monanchocidin A (MonA) is an alkaloid, initially isolated from the marine sponge *Monanchora pulchra* [48]. Dyshlovoy et al. demonstrated that MonA induces apoptosis in bladder and prostate cancer cells [49]. LMP was confirmed by release of cathepsin B into the extracellular space and disappearance of red fluorescence of acridine orange coupled with the appearance of green fluorescence. Non-malignant cells were less sensitive to MonA. Inhibitors of lysosomes and lysosomal enzymes were able to block the cytotoxic effects of MonA, further supporting the role of LMP in MonA-treated cells.

Triptolide (TPL), the active compound from the Chinese herb *Tripterygium wilfordii* Hook F, has been used in traditional Chinese medicine for over two centuries. TPL activates lysosomal-mediated apoptosis in MCF-7 breast cancer cells [50]. MCF-7 cells are a good model system to study lysosomal cell death because they lack caspase-3, a key pro-apoptotic executioner gene [51]. We have previously demonstrated in cell fractionation experiments that cytosolic levels of cathepsin B increase in triptolide-treated cells during early stages of apoptosis [50]. Owa et al. detected a shift from red fluorescence to green fluorescence in experimental cells stained with acridine orange [50]. These findings support the disruption of lysosomal membrane integrity by triptolide. We detected the subcellular localization of cathepsin B in the cytosol of MCF-7 cells via fluorescence microscopy in triptolide-treated cells [52]. In another report, TPL sensitized TRAIL-resistant pancreatic cancer cells to apoptosis by promoting LMP [53]. Taken together, these results demonstrate that TPL preferentially induces lysosomal disruption to target the death of cancer cells.

RDD648, an analog of the natural molecule riccardin D, was shown to exhibit anticancer activities in breast cancer by targeting lysosomes in vitro and in vivo [54]. RDD648 neutralized the acidic pH in lysosomes and induced lysosomal leakage. This finding suggests that this molecule behaves as a lysosomotropic agent. RDD648 facilitated STAT3 translocation to the nucleus, and this was involved in lysosomal-mediated cell death in breast cancer cells. The role of STAT3 in this lysosomal cell death was confirmed by the finding that inhibition of STAT3 ameliorated LMP. Nuclear STAT3 was observed to bind to TFEB, leading to partial loss of TFEB, which is required for lysosome turnover. These results may contribute to the design of treatments for breast cancers that express STAT3. A derivative of riccardin D triggered DNA damage via cathepsin B-mediated degradation and LMP in prostate cancer cells [55]. Another report showed that a derivative of riccardin D caused significant reduction of xenograft tumors, and this cell death was accompanied by LMP [56].

4-Deoxyraputindole C, also called compound S, a component extracted from the *Raputia praetermissa* plant, was shown to cause cell death and cell cycle arrest in cancer cell lines [57]. The authors showed that compound S was most active against Raji, a lymphoma cell lineage, and the death was accompanied by LMP and loss of mitochondrial membrane potential. Compound S also induced cell cycle arrest at G_0_ and G_2_. Furthermore, Vital and colleagues observed that this cell death was not abrogated by the caspase inhibitor. Taken together, these results suggest that compound S induces cell death in a caspase-independent manner.

Short-chain fatty acids (SCFAs) trigger cell death in colorectal cancer associated with lysosomal membrane permeabilization and mitochondrial dysfunction [58]. SCFAs are normally produced in colon cells by bacterial fermentation [59]. Gomes and coworkers demonstrated that transformed colonocytes are more susceptible to SCFAs compared to normal colonocytes. Propionibacteria produce SCFAs, mainly propionate and acetate, which induce apoptosis in colorectal cancer cells. Another study found that acetate-induced apoptosis in colorectal cancer cells was accompanied by LMP and cytosolic translocation of cathepsin D [60]. Further studies are warranted to determine if modulation of SCFAs can be used in the treatment or prevention of colon cancer.

In Geng et al. (2015), Icariside II (IS), a natural plant flavonoid, decreased the viability of HepG2 hepatoblastoma cells in a dose- and time-dependent manner [61]. Cell death was accompanied by LMP. Electron microscopy revealed autophagosome engulfment of IS-impaired lysosomes. An accumulation of the lysosomal marker protein LAMP1, which is an indicator of lysosomal membrane changes, was observed in the cytosol. Furthermore, acridine orange staining decreased with IS treatment, suggesting that the lysosomal membrane was damaged.

(-)-Epigallocatechin-3-gallate (EGCG) is the most extensively studied tea polyphenol for its anticancer function [62]. Zhang and coworkers demonstrated that EGCG-mediated cell death was caspase-independent and non-apoptotic. Furthermore, the authors showed EGCG triggered LMP and leakage of cathepsins into the cytosol [62]. Their study showed that this lysosomal cell death was mediated by reactive oxygen species (ROS). The overproduction of ROS is known to promote LMP [63,64].

Leelamine, a lipophilic diterpene amine phytochemical, is a natural compound extracted from the bark of pine trees and is a lysosomotropic agent [65] with antitumor properties. Leelamine has been shown to inhibit proliferation of and induce cytotoxicity in prostate and breast cancer cells [66,67]. Leelamine accumulates in lysosomes, thereby preventing the translocation of cholesterol into the cytosol, leaving unbound cholesterol unavailable for cancer cells’ activities [67]. This finding suggests that the anticancer properties of leelamine are due to its lysosomotropic properties.

Tubeimoside I (Tub) is a derivative of the Chinese medicinal plant of the Fritillaria genus that was identified by screening a chemical library of natural products [68,69]. Tub-treated lung cancer cells demonstrated excessive ROS production, which resulted in cytosolic release of cathepsin B [69]. Cathepsin B increase was confirmed by measuring the green fluorescence intensity in acridine orange-stained cells [69]. The cytosolic release of cathepsin B promoted upregulation of cytochrome C in cytosolic fractions, as detected by western blotting [69]. These results suggest that Tub kills lung cancer cells via LMP and precedes mitochondrial membrane potential changes.

Resveratrol (RSV), a phytochemical present in red fruits, peanuts, and grapes, possesses antioxidant and anticancer properties. RSV induces cell death in cervical cancer cells by increasing LMP and modulating the expression of p53 [70]. This study also detected decreased mitochondrial membrane potential downstream of the deregulation of the lysosomal membrane.

**Table 1 medicines-08-00069-t001:** Natural products with anticancer properties related to lysosomal membrane permeabilization.

Natural Product	Cells	Doses	Mechanism for LMP	Reference
Pinus radiata bark extract	MCF-7 breast cancer cells	65 μg/mL	Chelation of intracellular calcium and zinc	[43]
Omega 3 fatty acid docosahexaenoic acid	TIB-75 liver cancer cells	IC_50_ 28 μM	ROS and iron catalyzed reactions destabilize lysosomes	[44]
Monanchocidin A	Genitourinary cancer cells	50 μM	Extracellular release of cathepsin B	[49]
Triptolide	MCF-7 breast cancer cells	10 ng/mL	Lysosomotropic agent; cytosolic release of cathepsin B	[50,52]
RDD648	MCF-7 and HCC1428 breast cancer cells, prostate cancer cells, xenograft tumors	1–5 μM in vitro; 30 mg/kg in vivo	STAT3 binding to TFEB induces loss of TFEB (needed for lysosomal turnover)	[54]
4-Deoxyraputindole C	Lymphoma cells	53–56 μM	Decreased acridine orange concentration inside lysosomes	[57]
Short-chain fatty acids	Colon cancer cells	Not determined	Upregulation of LAMP-2 induces punctate structures in lysosomes	[58]
Icariside II	HepG2 hepatoblastoma cells	20–30 nM	Upregulation of cytosolic LAMP-1	[61]
(-)-Epigallocatechin-3-gallate	HepG2 hepatoblastoma and HeLa cervical cancer cells	60 μM	ROS-triggered LMP	[62]
Leelamine	MCF-7 breast & LnCAP prostate cancer cells	1–5 μM; Not Determined	Lysosomotropic agent; accumulates in lysosomes and disrupts cholesterol transport from lysosomes to cytosol	[66,67]
Tubeimoside I	Lung cancer cells	20 μM	ROS accumulation damages lysosomal membrane	[69]
Acetate	Colon cancer cells	70–220 mM	Cathepsin D cytosolic release	[60]
Resveratrol	Cervical cancer cells	150–250 μmol/L	Relocation of acridine orange from lysosome to cytosol	[70]

Abbreviations: LAMP-1, lysosome-associated membrane protein 1; LAMP-2, lysosome-associated membrane protein 2; ROS, reactive oxygen species; STAT3, signal transducer and activator of transcription 3; TFEB, transcription factor EB.

## 4. Conclusions

Rupture of lysosomal membranes promotes release of lysosomal proteases into the cytosol and apoptosis in cancer cells [17,70]. Figure 1 depicts putative mechanisms by which natural products induce lysosomal membrane disruption.

In this review, we proposed that lysosomal disruption may preferentially kill tumors treated with natural products, representing a potential novel therapeutic option against malignancies. Lysosomes are an interesting target in cancer cells because of their bigger size and frailty compared to healthy cells [4,5]. LMP is a promising mode of therapy for cancers that are resistant to chemotherapy, radiation, or hormone therapy, due to its distinct mode of action. Further studies are warranted to decipher the precise mechanisms by which natural products induce lysosomal cell death in cancer cells.

## Figures and Tables

**Figure 1 medicines-08-00069-f001:**
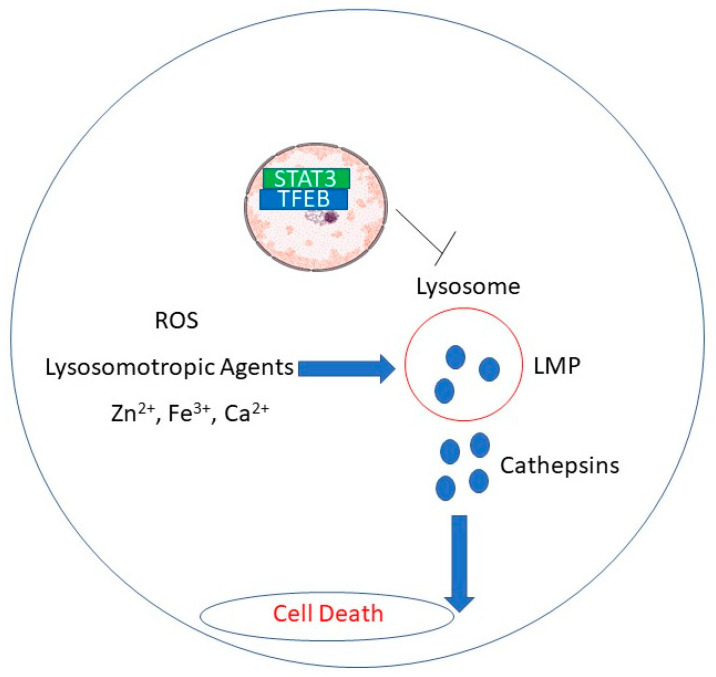
Putative mechanisms by which natural products induce LMP.

## Data Availability

Not applicable.

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
