# Peer review of "Natural Products Induce Lysosomal Membrane Permeabilization as an Anticancer Strategy"

_medicines, 2021, doi:10.3390/medicines8110069_

Round 1
Reviewer 1 Report
The review entitled "Natural Products Induce Lysosomal Membrane Permeabilization as an Anticancer Treatment" by Halaby is very well written and adresses an important piece of literature.
Authors may consider enlisting the executioner caspases the first time they mention it in the Review.
Author Response
Dear Reviewer, the revisions concerning the executioner caspases are indicated in red ink. Thank you for your helpful comments.
Reviewer 2 Report
The review is a good one. Author should recheck the literature to ensure Table 1 summarize all Natural products with anticancer properties related to lysosomal membrane permeabilization.
Author Response
Dear Reviewer,
I have included additional natural products that induce LMP in cancer cells. The edits are in red ink in the manuscript and in Table 1. Thank you very much for your helpful comments.